

# Supraglacial debris thickness variability: Impact on ablation and relation to terrain properties.

Lindsey I. Nicholson[1], Michael McCarthy[2,3], Hamish Pritchard[2] and Ian Willis[3]

[1] *Department of Atmospheric and Cryospheric Sciences, Universiät Innsbruck, Innsbruck, Austria.*

[2] *British Antarctic Survey, United Kingdom Research and Innovation, Madingley Road, Cambridge, UK*

[3] *Scott Polar Research Institute, University of Cambridge, Cambridge, UK*

*Correspondence: lindsey.nicholson@uibk.ac.at*





**ABSTRACT:** *Shallow ground penetrating radar (GPR) surveys are used to characterize the small-*
*scale spatial variability of supraglacial debris thickness on a Himalayan glacier. Debris thickness*
*varies widely over short spatial scales. Comparison across sites and glaciers suggests that the*
*skewness and kurtosis of the debris thickness frequency distribution decrease with increasing*
*mean debris thickness, and we hypothesise that this is related to the degree of gravitational*
*reworking the debris cover has undergone, and is therefore a proxy for the maturity of surface*
*debris covers. In the cases tested here, using a single mean debris thickness value instead of*
*accounting for the observed small-scale debris thickness variability underestimates modelled*
*midsummer sub-debris ablation rates by 11-30 %. While no simple relationship is found between*
*measured debris thickness and morphometric terrain parameters, analysis of the GPR data in*
*conjunction with high-resolution terrain models provides some insight to the processes of debris*
*gravitational reworking. Periodic sliding failure of the debris, rather than progressive mass*
*diffusion, appears to be the main process redistributing supraglacial debris. The incidence of*
*sliding is controlled by slope, aspect, upstream catchment area and debris thickness via their*
*impacts on predisposition to slope failure and meltwater availability at the debris-ice interface.*
*Slope stability modelling suggests that the percentage of the debris-covered glacier surface area*
*subject to debris instability can be considerable at glacier scale, indicating that up to 22% of the*
*debris covered area is susceptible to developing ablation hotspots associated with patches of*
*thinner debris.*





## 1. Introduction

Debris-covered glaciers are the dominant form of glaciation in the Himalaya (e.g. Kraaijenbrink et al. 2017), and are common in other tectonically active mountain ranges worldwide (Benn et al. 2003). Supraglacial debris cover alters the rate at which underlying ice melts in comparison to clean ice in a manner primarily governed by the thickness of the debris cover (e.g. Østrem, 1959; Loomis, 1970; Mattson et al., 1992; Kayastha et al. 2000; Nicholson and Benn, 2006; Reid and Brock, 2010): A thin supraglacial debris cover (< a few cm) enhances melt, while thicker debris cover reduces melt by insulating the ice beneath from surface energy receipts. Prevailing weather conditions, and local debris properties, such as albedo, lithology, texture and moisture content, also influence the amount of energy available for sub-debris ablation, and modify the exact relationship between debris thickness and ablation rate, but the general characteristics of the so-called Østrem curve are robust, further demonstrating the dominant role of debris thickness in this relationship (Fig. 1).

Both theory and observations indicate that the spatial variability of supraglacial debris thickness typically has both a systematic and a non-systematic component. Debris thickness tends to increase towards the glacier margins and terminus due to concentration by decelerating ice velocity, and increasing background meltout rate (e.g. Kirkbride, 2000). This systematic variation is evident in field measurements of debris cover thickness (e.g. Zhang et al., 2011), and in characterizations of debris thickness as a function of the surface temperature distribution observed from satellite imagery (e.g. Mihalcea et al. 2006; Mihalcea et al. 2008a; Mihalcea et al. 2008b; Foster et al. 2012; Rounce and McKinney, 2014; Schauwecker et al. 2015; Gibson et al. 2017). At local scales, debris thickness varies less systematically according to the input distribution, local meltout patterns and gravitational and meltwater reworking of the supraglacial debris. Manual excavations (e.g. Reid et al., 2012), observations of debris thickness made above exposed ice cliffs (e.g. Nicholson and Benn, 2012; Nicholson and Mertes 2017), and debris thickness surveyed by ground penetrating radar (McCarthy et al., 2017) demonstrate that debris thickness varies considerably over short horizontal distances. Thus, the thickness of debris over a sampled area of glacier surface is better expressed as a probability density function than a single value (e.g. Nicholson and Benn, 2012; Reid et al., 2012).

Exposed ice faces within debris-covered glacier ablation areas are known to contribute disproportionately to glacier ablation compared to their area (e.g. Sakai et al., 2000; Juen et al., 2014; Buri et al., 2016; Thompson et al., 2016), and it has been proposed that such 'ablation hotspots', along with stagnation, are the reasons for the observed similarity in surface lowering rates of otherwise comparable clean and debris-covered ice surfaces (e.g. Kääb et al., 2012, Nuimura et al., 2012). Given the strongly non-linear relationship between ablation rate and debris thickness (Fig. 1), patches of thinner debris within a generally thicker supraglacial debris cover can similarly be expected to contribute disproportionately to glacier ablation, but this has only rarely been considered (Reid et al., 2012). The implication of this would be that calculations of sub-debris ice ablation rate and meltwater production using spatially-averaged mean debris thickness may differ substantially from the actual meltwater generated from a debris layer of highly variable thickness within the same area. Therefore, there remains a critical need to be able to quantify not only mean supraglacial debris thickness, but also local debris thickness variability, in order to understand how debris cover is likely to impact glacier



behaviour, meltwater production and contribution to local hydrological resources and global sea level rise.

Meeting this need requires a better understanding of debris thickness variability and the controls upon it, ideally by means of more readily observable properties. Topographic data hasbeen used to predict soil thickness on hilly, extraglacial terrain under the assumption of steady state conditions (e.g. Pelletier and Rasmussen, 2009). However, associated soil thickness relationships as a function of slope curvature (Heimsmath et al., 2017) are based on progressive creep processes, while reworking of supraglacial debris cover occurs mainly as a result of gravitational instabilities such as 'topples, slides and flows' (Moore, 2017). Nevertheless, as the debris thickness that can be supported on a slope is related to slope angle, debris texture and saturation conditions (Moore, 2017) it might still be possible to find explicit relationships between topography and debris thickness. If high-resolution topography data, which is increasingly widely available, could be used to indicate local debris thickness variability, such information would complement spatially averaged mean supraglacial debris thickness values derived by other methods (cf. Arthern et al. 2006).

## 2. Aim of the study

This study investigates the evidence for small-scale debris thickness variability, assesses the impact of local debris thickness variability on calculated sub-debris ice ablation rates, and explores the potential for predicting local debris thickness variability from morphometric terrain parameters. First, debris thickness data from shallow ground penetrating radar surveys are used to characterize the small-scale spatial variability of debris thickness on a Himalayan glacier, examine evidence of gravitational reworking processes and compare the observed variability to previously published data. Second, the impact of the observed small-scale debris thickness variability on modelled sub-debris ablation rates is assessed. Third, a contemporaneous high resolution terrain model and optical imagery are employed to determine if the observed thickness variability can be predicted from more readily measured surface terrain properties. Finally, a slope stability model is calibrated with the GPR and ablation model data and used to determine the percentage of our study areas in the debris-covered ablation zone that are subject to debris instability, and potentially the formation of ablation hotspots, in mid-ablation season (August) conditions.

## 3. Study site and data

The Ngozumpa glacier is a large dendritic debris-covered glacier of the Eastern Himalaya, located in the upper Dudh Kosi catchment, Khumbu Himal, Nepal (Fig. 2a). The glacier has a total area of 61 km² of which the lower 22 km² is heavily debris-covered, with hummocky surface relief in the order of 50m over distances of 100m (Fig 2b), studded with supraglacial ponds and exposed ice cliffs (Benn et al., 2001). The NE and E branches are no longer connected dynamically to the main trunk (Thompson et al., 2016), which is fed solely by the W branch descending from the flanks of Cho Oyu (8188 m). The southernmost 6.5 km of the glacier is nearly stagnant (Quincey et al. 2009) and has a low surface slope of ~4°. The terrain of this glacier, its wasting processes and the evolution of surface lakes have been well studied through



a series of previous publications (Benn et al., 2000 & 2001; Thompson et al., 2012 & 2016), as
have the debris properties including limited measurements of debris thickness (Nicholson and
Benn, 2012).
Debris thickness over much of the debris-covered area is in excess of 1.0 m precluding
widespread manual excavation. However, in 2001 measurements of debris thicknesses exposed
above ice cliffs were made by theodolite survey at ~1 and 7 km from the terminus (Nicholson
and Benn, 2012). These data provided only coarse estimates of debris thickness as neither the
slope angle of the debris exposure, nor the impact of the theodolite bearing angle were
accounted for in the vertical offsetting used to obtain the debris thickness. In April 2016
terrestrial photogrammetry was used to create a high resolution scaled model of the local
glacier surface from which debris thickness estimates were made in a manner analogous to the
theodolite survey at a location ~2 km from the terminus near Gokyo village (Nicholson and
Mertes, 2017). At the same time, several GPR surveys, totalling 3301 m, were undertaken in this
area and a single 238 m GPR survey was done close to the glacier margin ~1 km from the glacier
terminus (Fig. 2a). Meteorological data are not available from the Ngozumpa glacier surface at
this site, so the ablation model was forced using several years of meteorological data measured
at the Pyramid weather station (27.95° N / 86.81°E, 5035 m a.s.l.) operated by the Ev-K2-CNR
consortium (http://www.evk2cnr.org/cms/en) in the neighbouring valley. A digital terrain
model generated from Pleiades tri-stereo imagery acquired in April 2016 is used to relate the
measured debris thicknesses to the glacier surface terrain.
**4.  Methods**
138        4.1  GPR debris thickness data collection and processing

GPR measurements were made between 31st March and 20th April 2016 broadly following the
methods of McCarthy et al. (2017). Debris thickness was sampled in 36 individual radar
transects, covering sloping and level terrain with coarse and fine surface material. The GPR
system was a dual frequency 200/600MHz IDS RIS One, mounted on a small plastic sled and
drawn along the surface. Data were collected to a Lenovo Thinkpad using the IDS K2 FastWave
software.  This system produces two simultaneous radargrams for each acquisition. The 200
and 600MHz antennas have separation distances of 0.230 m and 0.096 m respectively. Data
acquisition used a continuous step size, a time window of 100 ms and a digitization interval of
0.024 ns. The location of the GPR system was recorded simultaneously at 1 s intervals by a low
precision GPS integrated with the IDS which assigns a GPS location and time directly to every
twelfth GPR trace, and by a more accurate differential GPS (dGPS) system consisting of a
Trimble XH and Tornado antenna mounted on the GPR and a local base station of a Trimble
Geo7X and Zephyr antenna.
Radargrams were processed in REFLEXW (Sandmeier software) by applying the steps shown in
Table 1. The reflection at the ice surface was picked manually wherever it was clearly
identifiable and was not picked if it was indistinct. The appropriate signal velocity for the
supraglacial debris was obtained by burying a 1.5 m long steel bar to a known depth and then
passing the GPR over the buried target and picking the two-way travel time to its reflection (Fig.



3 a and b). Both fine and coarse material gave similar wave speeds (0.15 and 0.16 m ns$^{-1}$). These
were averaged to obtain a bulk value that is considered representative for all the radar lines
measured and is comparable to values from the debris-covered Lirung glacier, central Nepal
(McCarthy et al., 2017). Debris thickness was calculated using ice surface two-way travel times
and the mean of the two wave speed measurements (0.16 m ns$^{-1}$), taking the geometry of the
GPR system into account. Uncertainties were propagated according to McCarthy et al (2017)
and range from 0.14-0.83 m, generally increasing with debris thickness.
During processing, the integrated GPS locations (typical accuracy of ~ 3 m) were substituted for
dGPS locations (typical post-processed accuracy of < 0.05 m) by matching GPS and dGPS
timestamps. Where differential correction was not possible due to a lack of visible satellites, the
integrated GPS locations were used. The locations of GPR data collected between timestamps
were interpolated linearly in REFLEXW. Where the ice surface was identifiable in radargrams of
both frequencies, the measurement made using the higher frequency was assigned because
higher frequencies give higher precision. GPR data quality was assessed by comparing debris
thicknesses calculated using picks from the two different frequencies in the same location (Fig.
3c) and by comparing debris thicknesses at transect crossover points (Fig. 3d). In both cases,
points fit well to the 1:1 line. To show how debris thickness varies with topography, radargrams
were topographically corrected for display purposes after the ice interface had been picked.

175       4.2  Ablation modelling

In the absence of suitable field measurements of sub-debris ice ablation, a model of ice ablation
beneath a debris cover was applied to assess the impact of debris thickness variability on
calculated ablation rates. As recent, high quality, local meteorological data are not available to
force a time-evolving numerical model, typical ablation season conditions measured at the
nearby Pyramid weather station were used to force a steady-state model of sub-debris ice
ablation that has been previously published and evaluated against field data (Evatt et al., 2015).
Ice ablation conditions are generally restricted to the summer months in the eastern Nepalese
Himalaya (Wagnon et al., 2013). For the illustrative simulations performed here, the model was
forced with mean August meteorological conditions from 2003-2009 (<2% of August hourly
data are missing), and assuming the ice temperature to be 0°C. This provides forcing variables
of air temperature (3.27°C), incoming shortwave (208 Wm$^{-2}$) and longwave (314 Wm$^{-2}$)
radiation, wind speed (1.94 ms$^{-1}$) and relative humidity (97%). Appropriate debris properties
for dry debris in summer time on the Ngozumpa glacier were adopted from Nicholson and Benn
(2012), whereby debris properties of effective thermal conductivity, dry surface albedo and
porosity were taken to be 1.29 Wm$^{-1}$ K$^{-1}$, 0.2 and 0.3 respectively. Ice albedo, debris thermal
emissivity and the debris surface roughness length, friction velocity and exponential decay rate
of wind were adopted from Evatt et al. (2015).
The model is used to generate an Østrem curve and associated surface debris temperature for
the stated inputs, as a function of debris thickness. The model does not account for variability in
surface energy receipts due to local or surrounding terrain, or the effects of spatially or
temporally variable debris properties other than thickness, and the chosen input properties are
only approximate. However, this does not preclude its illustrative use in investigating the
influence of variable debris thickness on calculated ablation rate. Modelling was carried out for





three sites for which local debris thickness data is available: (i) the margin study area ~1km
from the glacier terminus, (ii) the main Gokyo study area ~2 km from the terminus, both
measured by GPR in 2016, and (iii) the upglacier study area ~7 km from the terminus,
measured by theodolite survey in 2001 (Fig 2). Ablation rate and surface temperature is
calculated for the mean debris thickness is compared to that yielded by multiplying the
percentage frequency distribution of debris thickness with the modelled Østrem and surface
temperature curves. Ablation totals for the month of August are calculated and that derived
using the mean debris thickness value is expressed as a percentage deviation of that derived
using locally variable debris thickness. Used in this form we assume the model itself to be error
free. To isolate the error associated with debris thickness, all other model inputs are also
assumed to be error free. Each GPR debris thickness has an associated error, but as no
quantified error assessment is available for the thickness values measured by theodolite at 7 km
from the terminus a fixed error of ± 0.15 m was applied to these data. The model was run with
maximum and minimum debris thickness values according to the assigned errors, to provide an
indication of uncertainty of the reported percentage difference in monthly total ablation.

### 4.3 Terrain analysis

In order to assess the static relationship between the debris distribution and terrain properties,
we used a 5 m resolution digital terrain model (DTM) derived from Pléiades optical tri-stereo
imagery taken during the field campaign on the 12th April 2016. The DTM was generated from
photogrammetric point clouds extracted from the Pléiades imagery, using a semi-global
matching (SGM) algorithm (Hirschmüller, 2008) within the IMAGINE photogrammetry suite of
ERDAS IMAGINE. The three images of each triplet were imported and the rational polynomial
coefficients (RPC) provided with the Pléiades data were used to define the initial functions for
transforming the sensor geometry to image geometry. With those transformation functions,
individual geometries of each image in the triplet were orientated relative to each other. To
obtain the most accurate exterior orientation possible, initial RPC functions were refined using
automatically-extracted tie points. The calculated point clouds were then filtered for outliers,
mainly found in very steep and shaded areas, using local topographic 3D filters applied in SAGA
GIS software, and converted into a 5 m-resolution DTM using the average elevation of all points
within one raster cell as the elevation value for the cell. Gaps were present in very steep areas,
where there was cloud, and in areas with low contrast because of fresh snow or liquid water.
Terrain properties were extracted using the ArcGIS tools Slope, Aspect and Curvature. GPR data
were resampled to the same resolution as these rasters (5 m) by taking the mean of the
measurements that occurred within each pixel. This was done using the Point to Raster tool in
ArcGIS. GPR data within 5 m of ice cliffs were excluded for comparisons made between debris
thickness and topography, in order that their slope, aspect and curvature were not
misrepresented. Similarly, GPR data for which dGPS locations were not available were excluded
due to their lack of positional accuracy.
Ponded water at the surface is associated with the deposition of layers of fine sediments and
rapid sedimentation by marginal slumping (Mertes et al., 2017). The recent history of ponded
water on the parts of the glacier surface sampled by the radar transects was mapped using air
photographs from 1984, and seven cloud-free optical satellite images spanning 2008-2016.




These images consisted of six Digital Globe images, one CNES/Astrium image, all obtained via
Google Earth, and the optical image from the 2016 Pleiades acquisition used to generate the
DTM.

### 4.4 Slope stability modelling and classification

Slope stability modeling was carried out following Moore (2017). For the three study areas
shown in Fig. 2, debris was classified as either stable or unstable. Unstable debris was further
classified as being unstable due to:
1.  Oversteepening, where surface slope exceeds the debris-ice interface friction coefficient,
2.  Saturation excess, where the modeled water table height is greater than the debris
thickness, and
3.  Meltwater weakening, where the modeled water table height is less than the debris
thickness, but debris pore pressures are sufficiently raised to cause instability.
Surface slope (see Section 4.3), modeled midsummer ablation rate (see Section 4.2), upstream
contributing area, and mean debris thickness (see Section 4.1) were used as inputs to the
model. Upstream contributing area was determined from the DTM in ArcGIS using the Flow
Direction and Flow Accumulation tools. Sinks in the DTM were filled if they were less than 3 m
deep, following Miles et al (2017), using the ArcGIS Sink and Fill tools. Surface water flowpaths
were also determined using the Stream To Feature tool.
The model also requires input values for the debris-ice interface friction coefficient, the
densities of water and wet debris, and the saturated hydraulic conductivity of the debris. A
value of 0.5 was used for the debris-ice interface friction coefficient, following Barrette and
Timco (2008) and Moore (2017). Values of 1000 and 2190 kg m$^{-3}$ were used for the densities of
water and wet debris, respectively, where wet debris was assumed to have a porosity of 0.3,
after Conway and Rasmussen (2000), and the density of rock was assumed to be 2700 kg m$^{-3}$
after Nicholson and Benn (2006). The saturated hydraulic conductivity of the debris, which is
the parameter around which there is most uncertainty, was determined using the GPR data.
Sections of the GPR transects, and subsequently their corresponding DTM pixels, were defined,
by visual inspection on the basis of the debris morphology, as either stable or unstable. Sections
of thin debris on steep slopes were considered to be unstable if they occurred among sections of
thick debris on shallow slopes. Sections of anything not considered to be unstable were
considered to be stable. Debris stability was then modeled for the same DTM pixels using a wide
range of conductivity values. The conductivity value that minimized the difference between the
number of pixels that were modeled and observed as being stable or unstable was considered to
be optimal. Minimization was carried out using ROC analysis, following Fawcett (2006) and
Herreid and Pellicciotti (2017). The resulting saturated hydraulic conductivity value of 40 m d$^{-1}$
is well within the expected range of $10^{-7}$-$10^3$ m d$^{-1}$ (Fetter, 1994), and is consistent with the
debris being well-drained.
The percentage areal coverage of debris instability was calculated for each of the three study
areas (Fig. 2). This was done both including and excluding ice cliffs and ponds, where ice cliffs
and ponds were manually digitized from the orthophoto associated with the DTM.





The GPR data, DTM and associated orthophoto were collected in March/April 2016, while slope
stability modeling was carried out using midsummer (August) ablation rates. It is likely that the
debris on a given slope becomes more or less stable seasonally with changes in ablation rates.
However, GPR observations of debris instability in March/April are likely to be representative
of midsummer debris instability for saturated hydraulic conductivity as maximum melt is
expected in midsummer. Similarly, while pond incidence and area vary seasonally on Himalayan
glaciers, recurrence rates are generally high (Miles et al., 2016), so manually digitized ponds and
ice cliffs for March/April are assumed to be broadly representative of ponds and ice cliffs in
midsummer for percentage area debris instability calculations excluding ponds and ice cliffs.
Finally, model results should be treated only as a best approximation because the model
assumes debris thickness and ablation rate are spatially homogeneous in each study area,
which, as discussed by Moore (2017), is clearly not the case.
**5.  Results and discussion**
*5.1  GPR debris thickness and variability*
The quality of the GPR data is generally high. The ice surface was clearly identifiable through the
debris in the majority of the radargrams collected. This is likely because the GPR system was
used in 'continuous-mode' and appropriate acquisition parameters were used. For those
radargrams in which the ice surface was not easily identifiable, the debris was generally too
thick. This means there is the possibility of a slight thin bias in the data. However, penetration
depth was often greater than 7 m, which is likely near the maximum debris thickness. Debris
thickness was found to be highly variable with a total range of 0.18 to 7.34 m (Fig. 4 and
examples in Fig 5). There is coherent structure to the debris thickness variation along transects
(Fig. 4): In some areas, changes in debris thickness along the transect are gradual, while in a
number of cases, there are abrupt changes in debris thickness along a transect associated with
pinning points or topographic hollows and cavities in the underlying ice, which the debris cover
fills (see Section 5.3 and Fig. 6).
Simple statistics of the debris thickness derived from the GPR samples of this study compared
with debris thickness datasets available from other glaciers are given in Table 2. Mean debris
thickness measured by GPR towards the glacier margin is thicker, and shows wider spread and
lower skewness and kurtosis, than the GPR thickness data collected at the Gokyo study area
(Table 2; Fig. 4; Fig 5a-c). The percentage frequency histogram of GPR debris thickness from the
glacier margin has a similar shape, but a positive offset compared to data obtained by surveying
of ice faces about 1 km from the glacier terminus in 2001, while the GPR data from Gokyo agrees
closely with the estimates of debris thickness from the photographic terrain model (Nicholson
and Mertes, 2017).  The 2001 surveyed debris thickness data from further upglacier (Nicholson
and Benn, 2012) is thinner, more skewed, and has higher kurtosis than the sites further
downglacier (Fig. 5a-c).
Clearly, while debris thickness shows small-scale variability in all cases on the Ngozumpa
glacier, the details of that variability differ from site to site. This is also observed when
considering data from other glaciers (Table 2; Fig. 5). Debris thickness at the Lirung glacier,
central Nepal shows a bimodal distribution not replicated at the other sites. This is suspected to



be due at least partly to sampling bias, as the measurements were made to test the GPR method rather than to characterize typical debris thickness at this glacier. At Suldenferner, in the Italian Alps, debris thickness measured across the whole debris-covered area by excavation, and along cross- and down-glacier transects by GPR, shows a substantially thinner mean than the Himalayan cases, with greater skewness and kurtosis. The debris cover on the medial moraine of Haut Glacier d'Arolla in the Swiss Alps is even thinner with yet more pronounced skewness and kurtosis. Thus, debris thickness variability at the Alpine sites shown here is more comparable to that of the upper Ngozumpa, while the Lirung glacier measurements appear broadly more similar to sites further downglacier on the Ngozumpa glacier.

The medial moraine on Haut Glacier d'Arolla emerged during glacial recession in the second half of the 20[th] century (Reid et al., 2012), offering an example of a recently developed debris cover. The debris-covered part of Suldenferner developed its continuous debris cover since the beginning of the 19[th] century, when the glacier was mapped with debris cover below ~2500 m and only surficial medial moraine bands extending up to 2700 m (Finsterwalder and Lagally, 1913). The Nepalese glaciers are thought to have been debris-covered for longer (Rowan, 2016), although it remains unclear when their debris covers first developed.

The percentage frequency distributions shown in Fig. 5, viewed in the context of the relative 'maturity' of the debris covers sampled, are suggestive of a progressive change in skewness and kurtosis debris thickness variability over time, as debris accumulates and undergoes progressively more gravitational reworking. The more mature debris covers on the Ngozumpa and Lirung glaciers is generally thick and characterised by hummocky terrain (cf. Fig. 2b), dissected with ponds and ice faces, whereas, the less mature debris cover on Suldenferner is generally thinner and the terrain is less hummocky, with relief primarily associated with incision by supraglacial streams. Similarly, the observed progressive change in thickness and skewness/kurtosis of the debris sites downglacier on the Ngozumpa glacier would reflect the downglacier increase in maturity of the debris covered surface.

### 5.2 Ablation modelling using mean and variable debris thickness

Ablation was calculated for three locations on the Ngozumpa glacier (Fig. 2) encompassing different mean debris thickness and debris thickness variability (Fig. 5; Fig. 6a), that might reflect different stages in debris cover maturity (see Section 5.1), but it should be noted that the sampling method and sample number differs between locations (Table 2).

The ablation calculated for typical August conditions using the mean debris thickness for each location on the glacier totalled 0.07, 0.11 and 0.32 m of ice surface lowering over the month at the 1, 2 and 7 km sites respectively. This agrees with the general expected patterns of ablation gradient reversal towards the terminus of a debris-covered glacier (e.g. Benn and Lehmkuhl, 2000; Bolch et al., 2008; Benn et al., 2017). Accounting for the percentage frequency distribution of debris thickness increased the monthly total surface lowering due to ablation to 0.08, 0.16 and 0.46 m, at 1, 3 and 7 km respectively. In these illustrative example, using a mean debris thickness instead of the local frequency distribution of debris thickness, underestimates the ablation rate at these sites by 11-30 % over a month of typical August conditions (Fig 6c). These values are specific to the cases presented here but can be considered indicative of the order or the effect of using mean debris thickness instead of the local variable debris thickness.





Considering the maximum and minimum error bounds of the debris thickness distribution (Fig
6a and c) increases the range of this underestimate to 10-40%. This suggests that local mean
debris thickness, and also other measures of central tendency (tested but not shown), are likely
to be poor metrics for ablation modelling for typical debris cover. Instead, sufficient data points
of debris thickness to capture the local variability are likely to give a more reliable ablation
estimate from model simulations. As the melt rate in the 'thin debris' part of the Østrem curve
responds more sensitively to changes in debris thickness than it does in the 'thick debris' part of
the curve, the impact of accounting for local spatial variability in debris thickness varies
inversely with debris thickness (Fig 6c). This is compounded by the fact that thinner debris
appears to have more skewness and kurtosis in the percentage frequency distribution of debris
thickness, meaning that the offset between the calculated mean debris thickness and the typical
debris thickness is likely to be greater.
Highly variable debris thickness can be expected to impact methods of mapping debris
thickness using thermal-band satellite imagery, as our data show that the debris thickness
variability within individual pixels of a thermal-band satellite image may be large. The modelled
surface temperature for mean August conditions was 19.5, 19.0 and 16.6°C for the mean debris
thickness at the margin, Gokyo and upglacier study areas respectively. Accounting for the local
debris variability at the lowest site altered the calculated surface temperature by < 0.1°C, and, at
the middle and upper locations, reduced the calculated surface temperatures by 0.5 and 1.5°C
respectively (Fig 6d). This highlights the manner in which variable debris thickness can be
expected to influence the pixel values in satellite thermal imagery, whereby a mean debris
thickness calculated from a pixel temperature can be expected to underestimate the true mean
debris thickness.
*5.3 Relationships between debris thickness and terrain properties*
Visual inspection of the radargrams indicates that the thinnest debris cover occurs on steep
slopes (Fig. 7a and b). On the basis that slope failure typically redistributes mass from areas of
high slope angle, and that debris sliding was often experienced while collecting the GPR data, it
seems likely that this is the result of high debris export rates in these areas due to frequent or
recent slope failure in the form of sliding events (c.f. Lawson, 1979, Heimsath et al. 2012). Here,
the debris surface is approximately parallel to the ice surface, and this appears to be a
characteristic of debris covers at or near the limits of gravitational instability. Localized areas of
thick debris are found below steep slope sections in the form of infilled ice-surface depressions.
Modelled surface flowpaths (Fig. 7b) cross-cut the GPR transects where these depressions are
located, indicating that they were likely incised by meltwater. This suggests that meltwater is
transported in sub-debris supraglacial channels (c.f. Miles et al. 2017), but also that meltwater
routing is a local control on debris thickness by providing topographic lows that become infilled
by debris. Additionally, it seems likely that meltwater channels undercut steep slopes, thereby
causing debris failure. Steep slopes on debris-covered glaciers are relatively short, so
undercutting would have the combined effect of increasing slope angle and also reducing the
confining force (or buttressing effect) imparted by down-slope debris cover. In some places,
thick debris is contained behind pinning points of the underlying ice (Fig. 7a and b), which
results in the occurrence of talus slopes (Fig. 7a), this stabilizes the debris and increases the





confining force. Thick debris on convex, divergent terrain provides evidence of topographic
inversion due to differential ablation (Fig. 7c).
The single glacier margin transect shows increasing debris thickness towards the glacier margin
(Fig. 4b and Fig. 7e). This is expected as a result of: (i) material delivered onto the glacier from
the inner flanks of the lateral moraines as they are progressively debuttressed by glacier surface
lowering; and (ii) lower surface velocities at the glacier margins, hence slower debris advection
rates. The Ngozumpa glacier and others in the region typically have troughs at the boundary
between the glacier and the lateral moraine, and evidence of thicker debris here reinforces the
idea that these troughs are eroded by meltwater routed along the glacier margins (Benn et al.,
416    2017).

Since 1984, the development of supraglacial ponds within the Gokyo study area is likely to have
affected two areas of radar transects: Several transects towards the north of the Gokyo study
area, which were partially affected by lakes in 2012 and 2014, and a single transect towards the
east of the Gokyo study area, which was partially affected by lakes in all the sampled years
except 2014 and 2016 (Fig. 4). One of the transects towards the north of the Gokyo study area
shows thick debris and some internal structures (Fig. 7e) including what may be a relict slump
structure, where a package of sediment fell into the lake from its margin as the lake expanded
(e.g. Mertes et al. 2016). Thick debris in former supraglacial lakes is likely due to high
sedimentation rates in the ponds and by slumping at lake margins during lake expansion
(Mertes et al. 2016). Modelling suggests that subaqueous sub-debris melt rates are low (Miles et
al. 2016), so debris thickening caused by the melt-out of englacial debris is likely to be minimal.
The radar stratigraphy over former lake beds suggests multiple near surface reflectors that can
reasonably be interpreted as fine lake sediments overlying coarser supraglacial diamict,
suggesting that the locally thicker sediments associated with lakes are due to deposition from
sediment-rich supraglacial and englacial meltwaters flowing into a more sluggishly circulating
pond.
The debris thickness sampled with GPR in this study does not show distinct relations with slope,
aspect or curvature ( Fig. 8a, b, c). Binning the thickness data with respect to slope indicates a
step decrease in debris thickness above surface slope angles of around 20-23˚ (Fig. 8a). This
may represent a transition from the low debris transport rates expected on low-gradient, stable
slopes, to the high-debris transport rates expected on steep, failure-prone slopes. While slope
and curvature are relatively evenly sampled by the dataset, the same is not true for aspect.
While southerly and north-easterly aspects are well sampled, samples are scarce in other aspect
sectors, rendering interpretation of potential aspect controls on debris thickness difficult (Fig
8e).  Tentatively, our data suggests thin debris is scarcer for northwesterly aspects, than others
(Fig. 8b, e). Comparing the GPR measurements with both slope and aspect simultaneously (Fig.
8e) shows what would be expected from Fig. 8a and 8b: That debris tends to be thicker on
northwest facing slopes, and thinner on steeper slopes away from the north-westerly sector.
During the pre-monsoon in the Himalaya, more melting is likely to occur on southeast-facing
slopes than southwest-facing slopes because clouds often reduce incoming shortwave radiation
in the afternoon (e.g. Kurosaki and Kimura, 2002; Bhatt and Nakamura, 2005, Shea et al., 2015).
This effect is observable in global radiation data (Fig. 8d).Distributing incoming shortwave
radiation on slopes of different slopes and aspects reveals the northwest sector to be the one



receiving least solar radiation in midsummer conditions(Fig. 8f). As a result slopes in this sector
may be expected to produce less meltwater meaning that debris water content, pore pressure
remain low, maintaining higher shear strength and greater stability, allowing thicker debris to
be sustained even on steep slopes (Moore, 2017). Samples from steep slopes in the south-east
sector are scarce, likely due to the higher melt rates resulting from higher solar radiation
receipts, serving to reduce slope angles here (Buri and Pellicotti, 2018). As a result of the
absence of steep slopes in the southeast sector, minimum debris thicknesses are displaced to
steeper slope angles flanking the aspect sector or highest midsummer solar radiation receipts.
No significant correlations were found between surface curvature and debris thickness (Fig. 8c),
but perhaps this is to expected, as the GPR samples only a snapshot of a dynamically evolving
surface.  Depending on the stage of topographic inversion sampled, thicker debris could be
found at the hummock summit or in the surrounding troughs. Furthermore, the predominance
of slope failure over slope creep mechanisms of gravitational reworking would serve to mask
any existing relationship with curvature.  Ultimately, it seems that the relationship between
debris thickness and morphometric terrain parameters (slope, aspect and curvature) is
complex.

*5.4 Slope stability modelling*

Slope stability modelling suggests that, under mid-August ablation conditions, the percentage of
the debris-covered area interpreted as potentially unstable for the three study areas of
Ngozumpa Glacier is between 13 and 34% including ponds and ice cliffs, and between 12 and
22% if ponds and ice cliffs are excluded (Fig. 9). The percentage of potentially unstable surface
area increases upglacier, as debris thickness decreases and ablation rates increase (Fig 6c).
Oversteepening was found to be the dominant cause of instability in all three study areas,
meaning that the debris is most likely to be unstable where surface slope is greater than ~27°
(i.e. greater than the inverse tangent of the debris-ice interface friction coefficient). In the Gokyo
and upglacier study areas, saturation excess was found to be the second most important cause
of instability and meltwater weakening the third. Here, it seems that the debris is thin enough
and ablation rates high enough for the debris to become saturated with surface meltwater. In
the downglacier margin study area, however, meltwater weakening was found to be more
important than saturation excess, presumably because the debris here is considerably thicker
and ablation rates providing meltwater are lower.
On the basis that thin debris is more likely to exist on unstable slopes, or on slopes that have
recently failed, and that debris-covered glaciers typically extend to lower elevations than
debris-free glaciers, these results have important implications for debris-covered glacier surface
mass balance. Debris gravitational instability provides a mechanism by which relatively large
parts of debris-covered glaciers can experience high melt rates, even if debris is generally thick.

## 6. Conclusions

Debris thickness is known to vary over the surfaces of debris-covered glaciers due to advection,
rockfall from valley sides, movement by meltwater, and slow cycles of topographic inversion.
The debris thickness data presented here suggest that the local debris thickness variability may




show characteristic changes in skewness and kurtosis associated with progressive thickening
and/or reworking of debris cover over time. On this basis the likely distribution of debris
thickness might be predicted by the maturity, or time elapsed since development, of the debris
cover found on a glacier surface.
For the thickly debris-covered glaciers of the Himalaya, sub-debris melt rates across the
ablation zones are generally considered to be small compared to sub-aerial melt rates at ice
cliffs (e.g. up to 5 cm d$^{-1}$, Watson et al. 2016) and sub-aqueous bare ice melt rates at supraglacial
lakes (e.g. 2-4 cm d$^{-1}$, Miles et al. 2016). Our GPR data confirm that the debris cover on
Ngozumpa Glacier is typically thick, with the thickest debris found on shallower slopes or the
sites of former supraglacial ponds. Here, the debris is too thick for the daily temperature wave
to penetrate to the ice (Nicholson and Benn, 2012). Consequently, even in core ablation season
conditions, typical melt rates are low across most of the debris covered area.  However,
processes of debris destabilization can form areas of thin debris within thicker debris. These
areas of thinner debris skew the spatially-averaged ablation rate in a manner that is analogous
to that caused by exposed ice faces. Here, sub-debris melt rates under thinner debris are
expected to be significantly above average, and even comparable with bare ice melt rates
further upglacier. We find that using mean debris thickness values in surface mass balance
models is likely to cause melt to be underestimated, and our results confirm previous
suggestions that debris thickness is better represented in surface mass balance models as a
probability density function (e.g. Nicholson and Benn, 2012; Reid et al., 2012).
On the surface of the Ngozumpa glacier, our data suggest that topography is an important
additional local control on debris thickness distribution, via slope and hydrological processes,
and also that  thick sediment deposits at the beds of former supraglacial ponds a are an
important additional control on the local variability of debris thickness. Surface debris appears
to be mobilized and transported by slope- and aspect-dependent sliding caused by sub-debris
melting, and most likely triggered by meltwater activity. Debris is redistributed from steep
slopes to shallow slopes and to ice-surface depressions that are often of hydrological origin.
However, the relationship between debris thickness and morphometric terrain parameters is
complex. While there is some apparent variation of debris thickness with slope and aspect,
whereby thinner debris caused by slope failure is more likely to occur on steeper slopes with
aspects that receive more abundant solar radiation, we find no meaningful variation with
curvature. This, combined with observations of slide-type debris morphology, suggests that
mass movement on the Ngozumpa glacier occurs on relatively short timescales and
predominantly by processes that occur at the limits of gravitational stability (e.g. Moore, 2017).
Slope stability modeling suggests that large areas of the glacier are potentially prone to failure,
and thus, as failure forms areas of thinner debris, that melting in these areas might be important
at the glacier scale.

*Data availability*  Debris thickness data measured on Ngozumpa glacier will be made publicly
available on https://zenodo.org/
*Author contribution* LN, MM and HP contributed to field data collection. LN analyzed the debris
thickness distributions, performed melt modelling and led the preparation of the manuscript.





MM, with guidance from HP and IW, processed the GPR data, performed terrain analysis, and
slope stability modelling. All authors contributed to finalizing the manuscript.
*Competing interests* The authors declare that they have no conflict of interest.
*Acknowledgements* This research is supported by the Austrian Science Fund (FWF) projects
V309 and P28521 and the Austrian Space Applications Program of the Austrian Research
promotion agency (FFG)  project 847999. M.M. is funded by NERC DTP grant number:
NE/L002507/1 and receives CASE funding from Reynolds International Ltd. HP was funded by a
British Antarctic Survey collaboration grant. The field team in Nepal was U Blumthaler, M
Chand, C del Gobbo, A Groos, A Lambrecht, C Mayer, H Pritchard, L Rieg and A Wirbel. C Klug
generated the DEM. Debris thicknesses data on Haut Glacier d'Arolla was collected by M
Carenzo, F Pelliciotti and  L Peterson and provided by T Reid.





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



*Table 1: Details of processing steps applied to radargrams, in order of use from left to right, using*
*REFLEXW software. T is the period of the transmitted signal, t is two-way travel time and f is*
*operating frequency.*

| operating frequency (MHz) | plateau declip | DC shift | dewow (ns) | align first breaks | timezero correct (s) | back-ground removal | band-pass filter | gain |
|---|---|---|---|---|---|---|---|---|
| 200 | whole profile | whole profile | 1.5T (7.5) | whole profile | $7.6719e^{-10}$ | whole profile | 0.25f, 0.5f, 1.5f, 3f | divergence compensation (scaling 0.1t) |
| 600 | | | 1.5T (7.5) | | $3.2022e^{-10}$ | | | |






Table 2: Statistics of sampled debris thickness variability measured at different locations on Ngozumpa, and other, glaciers by a range of methods.

| site | glacier | method | source | n | m | sample/m | mean | mode | skewness | kurtosis | 25% | 75% | min | max |
|------|---------|--------|--------|---|---|----------|------|------|----------|----------|-----|-----|-----|-----|
| Ngozumpa 1km | Ngozumpa | GPR* | this study (Margin) | 13983 | 238 | 58.75 | 3.33 | 2.19 | 0.48 | 1.84 | 2.23 | 4.35 | 1.74 | 5.96 |
| Ngozumpa 1km | Ngozumpa | theodolite | Nicholson and Benn, 2012 (upper) | 92 | 460 | 0.20 | 1.65 | 1.87 | 0.87 | 3.76 | 1.05 | 2.14 | 0.12 | 4.36 |
| Ngozumpa 3km | Ngozumpa | GPR* | this study (Gokyo) | 130926 | 3301 | 39.66 | 1.95 | 1.33 | 1.06 | 3.60 | 0.93 | 2.71 | 0.18 | 7.34 |
| Ngozumpa 3km | Ngozumpa | SfM-MVS | Nicholson and Mertes, 2017 | 1011 | 980 | 1.00 | 1.82 | 0.75 | 1.33 | 4.13 | 0.73 | 2.46 | 0.02 | 7.62 |
| Ngozumpa 7km | Ngozumpa | theodolite* | Nicholson and Benn, 2012 (lower) | 143 | 715 | 0.20 | 0.59 | 0.09 | 1.93 | 8.27 | 0.25 | 0.92 | 0.09 | 3.22 |
| Lirung | Lirung | GPR points | McCarthy and others, 2017 | 6198 | 354 | 17.51 | 0.66 | 0.39 | 1.07 | 3.24 | 0.32 | 0.93 | 0.11 | 2.30 |
| Suldenferner | Sulden | GPR | del Gobbo, 2017 | 61136 | 1000 | 61.14 | 0.32 | 0.29 | 0.07 | 3.39 | 0.26 | 0.38 | 0.00 | 0.74 |
| Suldenferner | Sulden | excavation | del Gobbo, 2017 | 101 | 10100 | 0.01 | 0.14 | 0.10 | 2.05 | 7.49 | 0.06 | 0.16 | 0.00 | 0.67 |
| Arolla | Arolla | excavation | Reid and others, 2012 ‡ | 488 | 976 | 0.50 | 0.07 | 0.01 | 6.29 | 68.86 | 0.02 | 0.08 | 0.01 | 1.50 |

* Data used in ablation modelling in this study

‡ Data from the medial moraine only, excluding measurements of patchy debris (< 0.01 m in thickness)





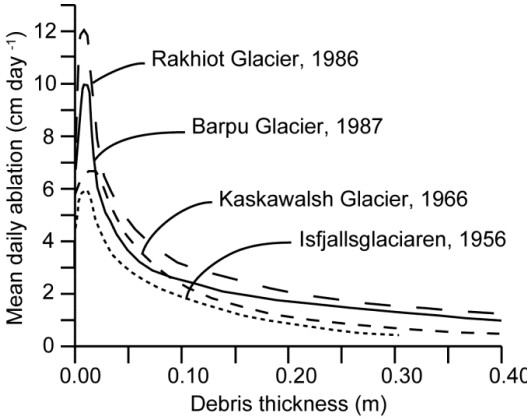

Fig. 1: Examples of the relationships between supraglacial debris thickness and underlying ice ablation rate at different glacier sites, redrawn from Mattson et al. (1993). The exact form of this relationship at each site varies with prevailing meteorological conditions and debris properties, but its general character is preserved.



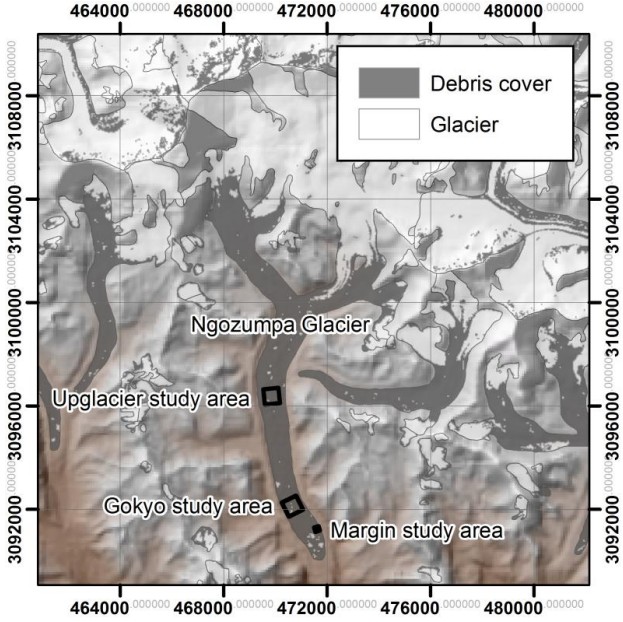

Fig. 2: (a) Ngozumpa glacier showing the key study areas, ~7, 2 and 1 km from the glacier terminus (b) Photograph showing example hummocky terrain in the upglacier study area – note the people for scale in the bottom right corner. Photo credit H. Pritchard.





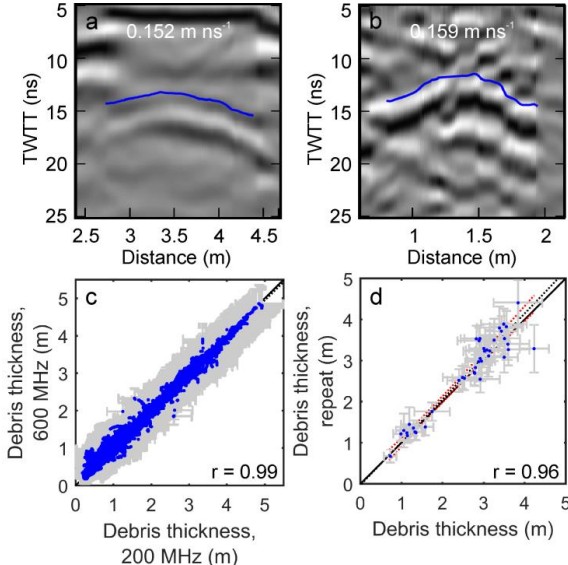

Fig. 3: Reflector used to identify signal velocity on Ngozumpa glacier in (a) fine-grained sediments and (b) coarse-grained sediments. Comparison of picked debris ice interface depths sampled simultaneously with different frequencies (c) and at transect intersection points (d).





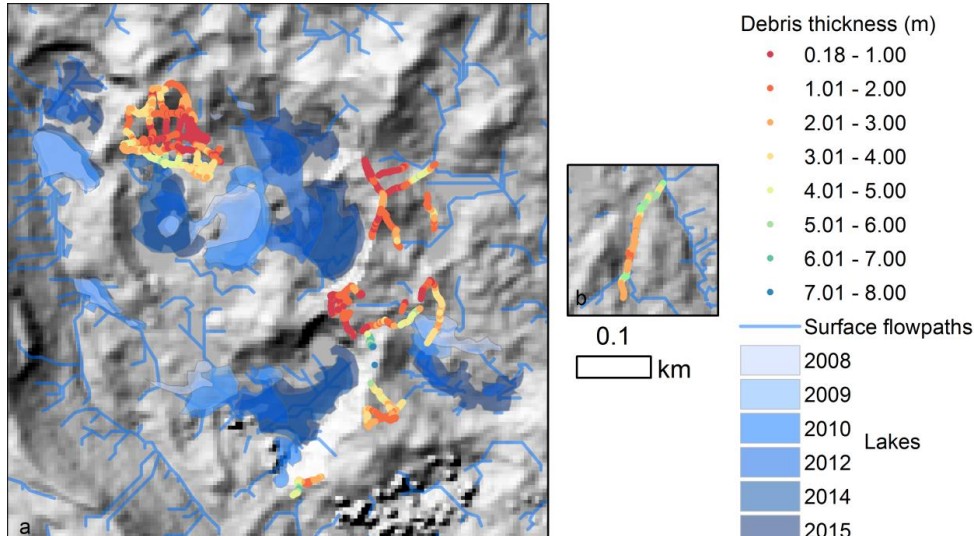

Fig. 4: Overview map of GPR debris thickness sampled on Ngozumpa glacier in 2016 overlain on
the hillshade from the Pleiades DTM , recent surface pond evolution, and surface flow paths for the
Gokyo(a) and Margin (b) study areas (Fig. 2).



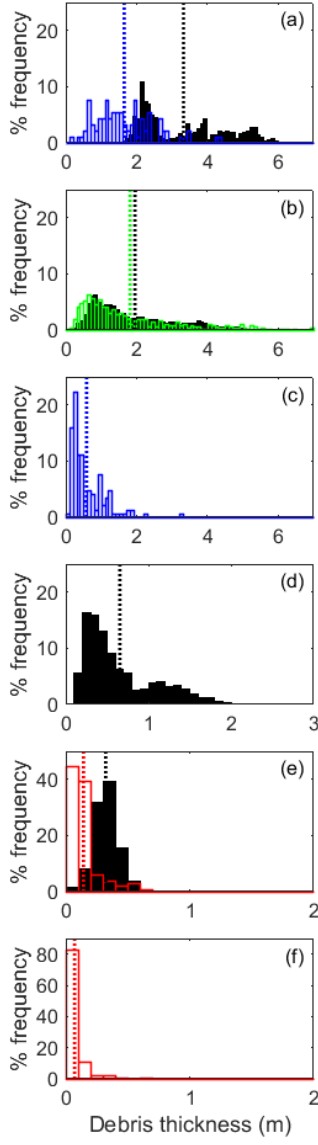

Fig. 5: *Percentage frequency histograms of debris thickness (hd) in 0.05 m intervals at (a) the lower Ngozumpa about 1 km from the terminus; (b) Gokyo area of Ngozumpa, about 2 km from the terminus; (c) upper Ngozumpa, about 7 km from the terminus; (d) over the lower tongue of Lirung glacier in central Nepal; (e) across the debris covered ablation area of Suldenferner/Ghiacciaio de Solda in the Italian Alps; (f) the medial moraine of Haut Glacier d'Arolla in the Swiss Alps. Measurement methods are GPR (black); theodolite surveys (blue); Structure from Motion (SfM-MVS) photographic terrain model (green) and excavation of pits (red). Note that axes vary between sites, and summary statistics of these distributions are in Table 2.*

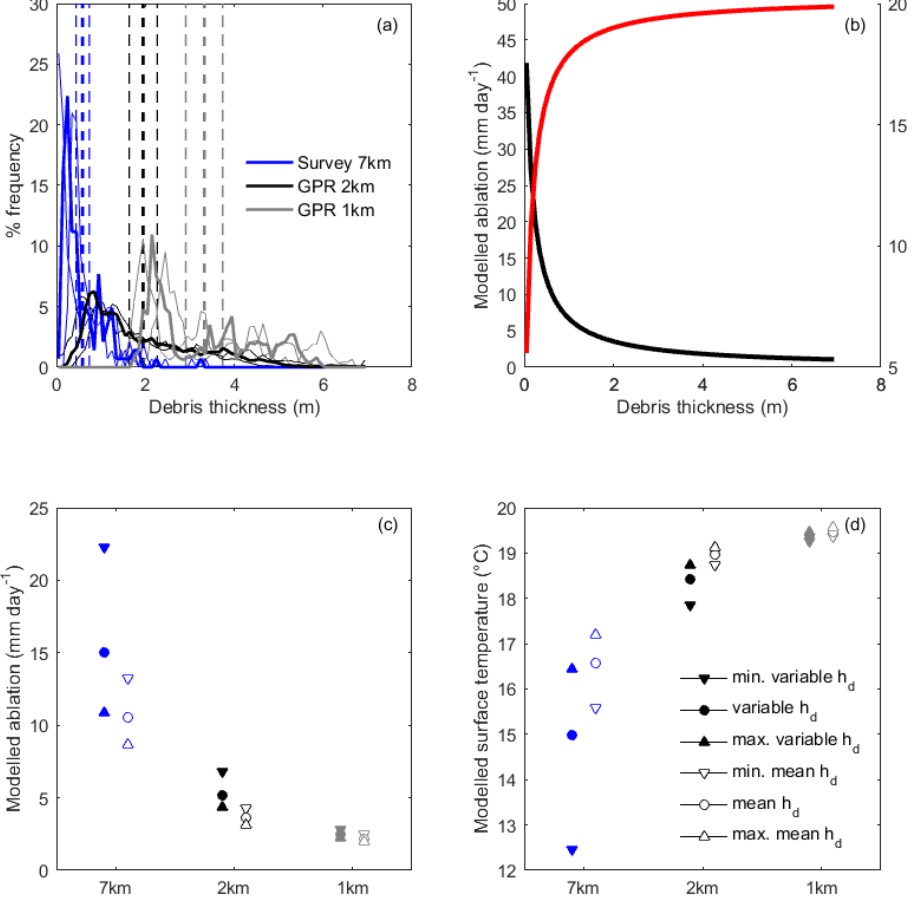

*Fig. 6: (a) Percentage frequency distributions from three locations on Ngozumpa glacier, showing the mean debris thickness at each site in dotted vertical lines: 3.33, 1.95 and 0.59 m thick respectively at 1, 2 and 7 km from the terminus. Thinner lines show the values for the maximum and minimum debris thickness conditions calculated from the limits of the individual debris thickness errors. (b) Modelled Østrem curve and surface temperature for mean August conditions. (c) Comparison of modelled ablation for different representations of the debris thickness at each site. (d) Comparison of modelled surface temperature for different representations of the debris thickness at each site.*




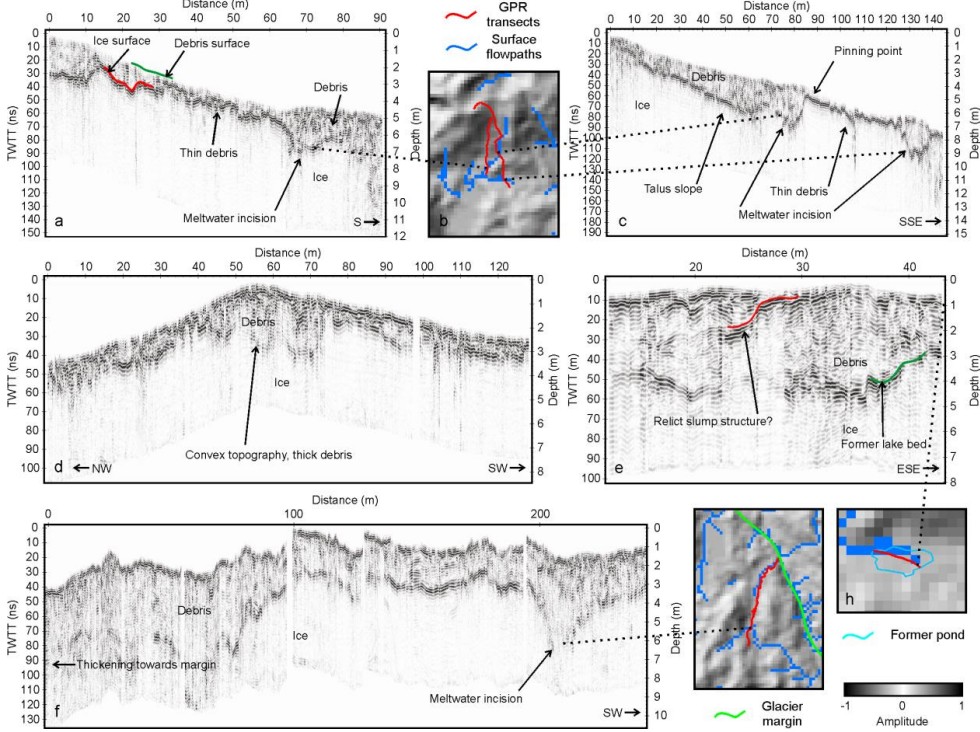

*Fig. 7: Example radargrams showing debris thickness variability and internal structures in relation to local topography and surface meltwater flow pathways.*





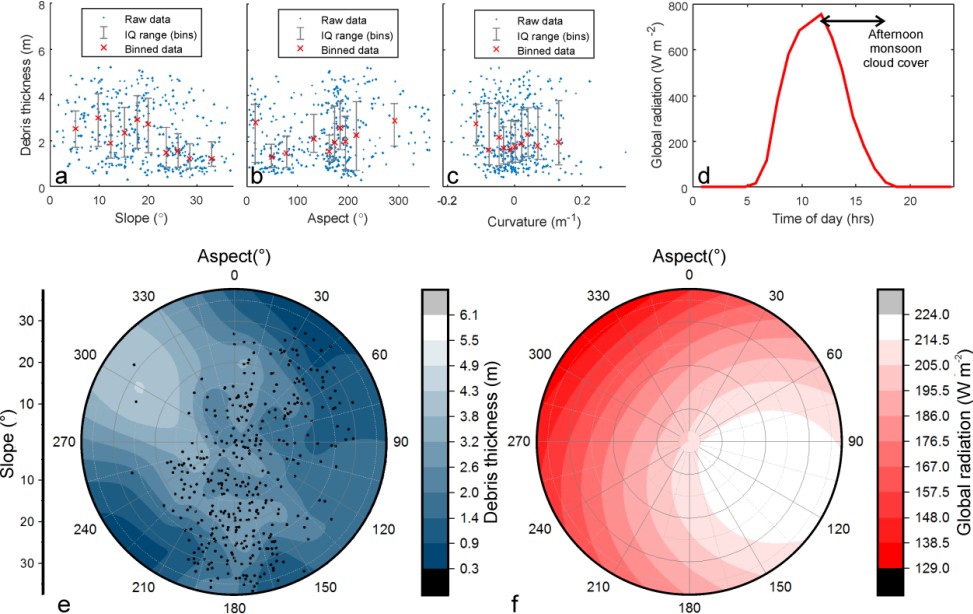

Fig. 8: Summary of relationships between measured debris thickness and terrain properties: (a)
debris thickness related to local slope angle; (b) debris thickness related to local slope aspect; (c)
debris thickness related to curvature (d) August global radiation data collected on the glacier
during the survey period; (e) hemispheric plot of debris thickness (showing sub-sampled data
points) related to slope angle and aspect; (f) hemisphere plot of August global radiation,
distributed on surfaces of different slope and aspect following Hock and Noezli (1997).



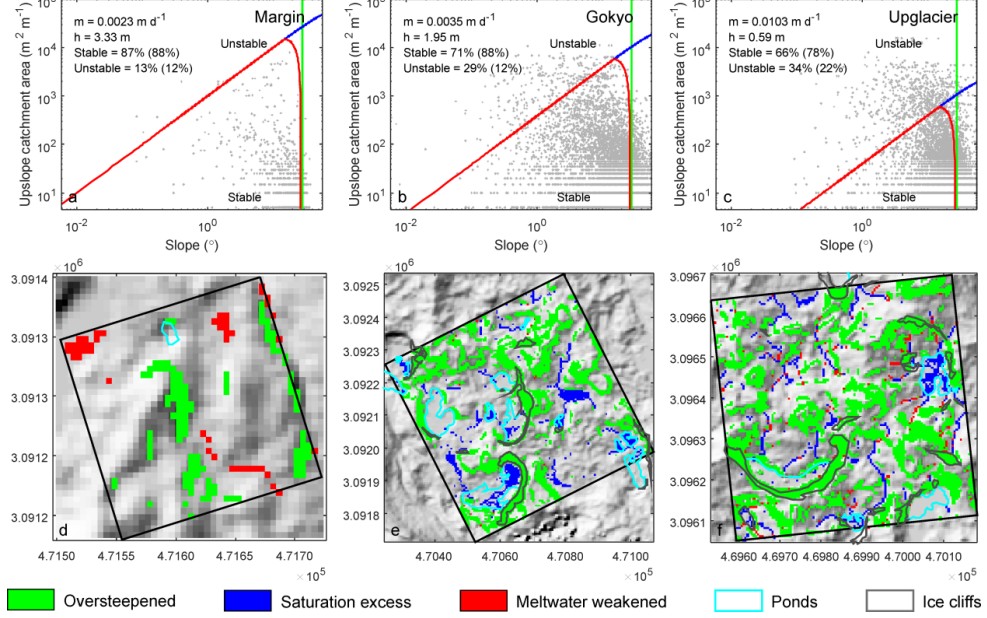

*Figure 9: Results of debris stability modelling: Upslope catchment area as a function of slope angle for the three study areas (a-c); points falling above or to the right of the plotted lines are unstable. Percentage area stability/instability values are given with lakes and ice cliffs included, and in brackets with lakes and ice cliffs excluded. Maps of spatial distribution of terrain stability classifications for each study area (d-e), highlighting ponds and ice cliffs.*