# Peer review of "Supraglacial debris thickness variability: Impact on ablation and relation to terrain properties."

_The Cryosphere, 2018_

## Referee Comment (RC1) · P. Moore (Referee) · 15 Jun 2018

GENERAL COMMENTS: Nicholson and co-authors report on the collection and analysis of surface topography and debris thickness data from selected sites across a debris-covered glacier margin in the Eastern Himalaya. Their data set is rich and valuable, and the analysis is reasonable and brings out many interesting features in the data. Two salient implications arising from their work are: 1) ablation rates estimated using mean debris thickness tend to underestimate ablation compared with methods that account for the frequency distribution of debris thickness; and 2) gravitational reworking of supraglacial debris plays a significant role in setting the frequency distribution of debris

thickness. Both of these implications are important for understanding and modeling the role of debris cover in glacier ablation, and for this reason, I think this paper could be an impactful contribution to the literature on debris-covered glaciers. However, in motivating the work and discussing these implications, I think there are some opportunities missed and issues not addressed that could make the paper more robust. For these reasons, I recommend minor to moderate revisions prior to acceptance.

SPECIFIC COMMENTS: 1. There is much discussion of descriptive properties of the debris thickness frequency distributions, particularly their skewness and kurtosis. These are certainly reasonable statistics to derive from populations of thickness measurements, and I can maybe infer their utility for this problem in retrospect. But it is difficult to understand from the introduction and methods why these statistics are reported in so much depth and what meaning the authors expect to convey through them about the processes on the glacier surface or the impacts on ablation. I suspect that (assuming a linear debris thickness axis) the thickness distribution is usually positively skewed in most glaciers, but is there a physical reason to expect that? Under what circumstances, if any, might we expect a negative skew? What could be the physical meaning of a low kurtosis – could we reason that more transport by gravitational failure would result in lower kurtosis? Would predominantly "diffusive" (slope-dependent only, as in landscape evolution modeling) transport result in higher kurtosis? What if gravitational instability were very widespread – would we expect multimodal distributions, with many thin areas that recently destabilized and also many thick areas that received slides/flows from above? Perhaps we don't know enough to address these questions in a formal if-then kind of way, but I do think it would help to build the case for caring about the shape of the distribution, and feel that some practical or theoretical basis should be offered to make the case.

2. This is partly a comment about insufficient information, but perhaps also a missed opportunity to elaborate on the use of frequency distributions to scale ablation rates from Ostrem curves. (Lines 202-205; 358-362). Their approach is described briefly

in lines 202-205, but it took several read-throughs for me to understand what the authors meant in that sentence. This could be made much clearer, and if this approach is widely used (it is entirely possible that I've overlooked something elsewhere in the literature) it might be nice to reference others who have used the method. More significant in my mind—though it would be reasonable to argue that this is beyond the scope of the work presented—I think an important link between these data and some powerful and more generalizable implications could be drawn here, but is missing. The authors are implicitly using the frequency distribution of debris thickness multiplied by the Ostrem curve as their "best" value for ablation modeling; otherwise they would have said that this method overestimates ablation compared to using the mean, rather than the converse. If they (as I suggest in #1 above) established some reasonable expectations or hypotheses about the shape of the debris thickness distribution, it should be possible to offer concrete hypotheses about how these distributions would affect ablation. Without too much difficulty I think one could establish an a priori expectation that a realistic frequency distribution of debris thicknesses would result in more predicted ablation than if the whole area had the mean of the same distribution. With that as a sort of hypothesis, the significance of a set of measured debris thickness distributions would be more apparent to the reader. I think that if these suggestions were implemented, this paper would read more clearly and could have a more substantial impact. As it is right now, there seems to be only a weak connection forged between the efforts at characterizing debris thickness distribution and patterns and the efforts to estimate ablation rates, and this connection is apparent (at least to me) only after reading the whole paper.

TECHNICAL CORRECTIONS (L = line in manuscript): L77. Add space between "has" and "been". L202-203. "Ablation rate and surface temperature [delete 'is'] calculated for. . ." L287. It is not clear what the "recurrence rates" refer to. Is this the repeated appearance of supraglacial ponds in a particular area? If so, is it distinct pond bodies, draining/refilling of unchanging basins, duration of ponding, or something else? L341. There seems to be a word like "of" missing between "kurtosis" and "debris". L364.

Change "order or the effect" to "order of the effect". L367-368. One could argue that since the use of mean debris thickness seems to consistently underestimate composite ablation rates, it is not worthless but can still have value as a lower bound. Going one step further, even the debris thickness distribution derived from higher-spatial-resolution measurements could include some spatial averaging, so at what point are we looking at a small enough area that refining the resolution even more wouldn't further increase ablation? L499 and L517. A nitpicky stylistic thing, but I dislike the word "shallow" used as the opposite of "steep". I suggest "gentle" or "gradual" instead. Section 5.4. I think the gravitational stability modeling is a reasonable piece to include in the analysis, but it would be prudent to present some assessment of the sensitivity of the model results presented (i.e., areal extent of predicted instability) to unknown values introduced to or inferred from the model, like the ice-debris friction coefficient or debris hydraulic conductivity. These could significantly change the results. Figure 2's caption and the text on line 109 indicate that there should be a panel (b) for Figure 2, but none appears in the copy of the manuscript I've seen.

---

## Referee Comment (RC2) · Anonymous Referee #2 · 12 Jul 2018

Authors have interesting story in this paper. They focused on the thin debris zone as 'ablation hotspot' at debris-covered glaciers. They evaluated the effect of 'ablation hotspot' when we estimate ablation under the debris layer based on the observed debris thickness frequency. Estimation of debris thickness have been tried using surface terrain properties. And they concluded that 'ablation hotspot' are possible to appear from the analysis of debris stability. There are several kinds of biases to estimate ablation of debris-covered glaciers using satellite thermal images. And evaluation of thin debris zone ('ablation hot spot') is one of the most significant factor to cause bias. I think this paper have demonstrated important issue to estimate ablation of debris-covered glaciers using satellite images. I have some main and specific comments as

follows.

<Main comments>

1) Author have analyzed debris thickness frequency which are measured by GPR, theodolite, SfM-MVS and excavation. I can imagine point measurements of debris thickness were carried out by theodolite, SfM-MVS and excavation. But, this paper have no information about the GPR. GPR also measure DT at each point or at some ranges in horizontal ( depend on the debris thickness??)?

2) You have classified stable and unstable debris-covered area at just 27 degree in slope gradient. But, usually, steep slope have low accuracy in satellite- or SfM-DTM. Margin site might have no oversteepened grids if you change the critical value of slope.

<Specific comments>

Please consolidate the location name. Even all expressions are summarized in Line199-201, different expressions at each figure are not easy to understand for readers. You have wrote 'Gokyo' and 'Margin' in Fig. 2, 4, 6c, 6d and 9, but expressed by distance from terminus in Fig. 5. Please check other part in the manuscript. 'lower' 'middle' and 'upper' were also found in Line 382.

L120 Please add information of altitudes at each three site.

L163 Author have wrote that 'McCarthy et al (2017) and range from 0.14-0.83 m, generally increasing with debris thickness' How much minimum thickness can be detect by GPR? Author have depicted Fig. 5, percentage frequency histgrams of debris thickness in 0.05 m. Thin debris are important for following analysis.

L175 '4.2 Ablation modelling' » It seems that temperature difference due to elevation difference at each three site have not been considered in this calculation. I think it is not necessary to consider the temperature difference, because the target of this calculation is to indicate the effect of debris thickness variability on ablation. But, you have to write that you have assumed that temperature were same with the Pyramids (?) at all sites.

L239-240 There are no information of data source of air photographs taken in 1984.

L299-301 'the debris was generally too thick. This means there is the possibility of a slight thin bias in the data. However, penetration depth was often greater than 7 m, which is likely near the maximum debris thickness.' » Those sentences have been written subjectively without support information. Authors cannot declare without some references.

L307 'see Section 5.3 and Fig. 6' » 'see Section 5.3 and Fig. 7'?

L343 I cannot find Fig. 2b.

L354 If you have assumed that temperature difference depending on the altitude were not considered for the estimate of ablation, you have to add the information here. If you take into account the temperature difference, you have to add altitude at each three sites and temperature lapse rate.

L355 and 362 The unit of vertical axis in Fig. 6c were mm day-1. but, 'm' in the manuscript. If both values indicate same things (I believe this), please consolidate. This figure and calculated values are very important in this paper.

L360 ' 1, 3 and 7 km respectively.' »' 1, 2 and 7 km respectively.' ???

L370-373 These sentence indicates very significant things to estimate ablation under debris-layer based on mean debris thickness. I recommend that each percent of debris thickness frequency between 0-0.5 m in debris thickness and each calculated ablation ratio between 0-0.5 m in debris thickness at each three site should be shown in the text.

L389 'Visual inspection of the radargrams indicates that the thinnest debris cover occurs on steep slopes (Fig. 7a and b).' » I cannot agree with this sentence. For me, by visual inspection, it seems that depression of ice surface are filled with debris, as a result, debris surface have flatter features than that of ice surface under the debris. All example show such tendency in Fig. 7.

Line394 'the debris surface is approximately parallel to the ice surface,' » This sentence does not conflict with previous sentence 'Visual inspection of the radargrams indicates that the thinnest debris cover occurs on steep slopes' at Line 389 ??

L390 '(Fig. 7a and b).'» '(Fig. 7a and c).'???

L397 'Modelled surface flowpaths (Fig. 7b) cross-cut the GPR transects where these depressions are located, indicating that they were likely incised by meltwater.' » I'm confused when I checked the cross section of Fig. 7c and f. You have calculated the flowpath based on DEM from Pleiades. But, the surface features indicated by GPR cross section do not represent cross section of flowpath (no depression). The surface elevation of debris is not true?

L417 and Fig. 7h » How did you detect the location of former pond? 'Former pond' in Fig. 7h means pond in 1984 ?

L434-435 'Binning the thickness data with respect to slope indicates a step decrease in debris thickness above surface slope angles of around 20-23 Íe (Fig. 8a).' » I recommend that difference between debris thickness at steep slope and those at gentle slope were significant or not statistically.

L434- As I wrote at Line 394, it seems that depression of ice surface are filled with debris, as a result, debris surface have flatter features than that of ice surface under the debris from Fig. 7 by visual inspection. Then, ice surface (not debris surface) curvature might have relation with debris-thickness. Probably I think some relation can be found between ice surface curvature and debris-thickness by optimizing the scale of curvature. But, it's not your purpose, and the analysis is limited at the GPR survey lines.

L448 'This effect is observable in global radiation data (Fig. 8d).'» If you add calculated solar radiation at the top of the atmosphere in Fig. 8d, it is easy to understand.

Fig. 4 Shore lines of ponds are unclear for me because they are overlapping. I recommend that only shore lines are depicted in this figure. And add the information of the background images.

Fig. 5 There is no information about the intervals of debris thickness in d-f. And please add the information that 'Dashed vertical lines indicate mean debris thickness' or something.

Fig. 7 Locations of Fig. 7b, f, h are required.

Table 2 I recommend that percentage of thin debris thickness frequency are necessary in Table 2. For example, the range between 0-0.25 m and 0.25-0.5 m in debris thickness. Because, the thin debris layer effect the bias of ablation and surface temperature.

---

## Author Comment (AC1) · 7 Sep 2018

**Response to reviewer 1:**

We would like to thank Peter Moore for his careful review and helpful comments. We hope we have addressed them adequately and as a result created a more impactful version of our manuscript.

We address these as follows:

Our responses are given in blue below and highlighted in the revised manuscript using track changes.

**Specific comments:**

These comments are broadly about improving the structure of the article in order to make it more impactful. This falls under two main categories:
1) The need for further explanation of the relevance of the debris distribution examples shown to allow the reader to ore clearly see the significance of these
2) Make some more generalized and hypotheses regarding the controls on likely debris thickness frequency distributions
3) Provide more concrete hypotheses regarding how ablation calculated using a given frequency distribution of debris thickness might be expected to differ from that calculated using mean debris thickness.

We address these as follows:

(a) Introducing some more speculative discussion of expected debris thickness distributions in the introductory section

"The limited available data shows the probability density functions or frequency distribution of debris thickness at a glacier or local scale to show varying degrees of kurtosis and typically a positive skew (e.g. Reid et al., 2012; Nicholson and Benn 2012), but the degree to which the frequency distribution deviates from normal, and the controls on the degree of kurtosis and skewness have not been well investigated. Nevertheless, some postulations can be made based upon the systematic and non-systematic variability components described above. As thick debris cover tends to form where there is little to no ice flux it follows that glaciers close to steady state will tend to be dominated by thin debris, causing the debris thickness frequency distribution to have a positive skew, while this might be expected to be less pronounced in sluggish debris-covered glacier termini, or even have a negatively skewed distribution on stagnant glacier tongues or rock glaciers, where ice flux is minimal. Glaciers with patchy debris at the surface are also more likely to have a positively skewed debris thickness distribution than continuously covered glacier surfaces due to gradual topographic inversion and lateral dispersal of debris from localised surface deposits (Anderson, 2000; Kirkbride and Deline 2013). Gently sloping smooth surfaced debris covered glaciers might be expected to experience less gravitational sliding than steeper or more chaotic glacier surfaces, and less gravitational reworking may favour relatively higher kurtosis than at sites where sliding and slope failures are common, and the frequency distribution of debris thickness can be rapidly reworked and potentially even develop multimodal distributions with many areas of thin, recently destabilized debris and also many areas of thick debris where material from slope failures has accumulated."

(b) More clearly describing the link between the debris thickness distributions and the Østrem curve

[revised manuscript text omitted]

**Technical corrections:**

L77. Add space between "has" and "been".

Done.

L202-203. "Ablation rate and surface temperature [delete 'is'] calculated for…"

Done.

L287. It is not clear what the "recurrence rates" refer to. Is this the repeated appearance of supraglacial ponds in a particular area? If so, is it distinct pond bodies, draining/refilling of unchanging basins, duration of ponding, or something else?

Changed 'recurrence rates are generally high' to 'seasonal ponds commonly reform at the same sites'.

L341. There seems to be a word like "of" missing between "kurtosis" and "debris".

Added 'of'.

L364.Change "order or the effect" to "order of the effect".

Done.

L367-368. One could argue that since the use of mean debris thickness seems to consistently underestimate composite ablation rates, it is not worthless but can still

have value as a lower bound. Going one step further, even the debris thickness distribution derived from higher-spatial resolution measurements could include some spatial averaging, so at what point are we looking at a small enough area that refining the resolution even more wouldn't further increase ablation?

We did not really intend to label the mean debris thickness as worthless, but highlight its potential limitations. Therefore we now write: "This suggests that while modelled ablation using local mean debris thickness can provide a lower bound this and other measures of central tendency (tested but not shown here), are likely to be poor inputs for ablation modelling for typical debris cover."

The issue of defining an 'appropriate' resolution of measurements is a good one to raise and remains a little problematic. We now say that: " ... sufficient data points of debris thickness to capture the local variability are likely to give a more reliable ablation estimate from model simulations." Although this only partially addresses the problem.

L499 and L517. A nitpicky stylistic thing, but I dislike the word "shallow" used as the opposite of "steep". I suggest "gentle" or "gradual" instead.

Changed 'shallow/er' to 'gentle/more gentle' throughout.

Section 5.4. I think the gravitational stability modeling is a reasonable piece to include in the analysis, but it would be prudent to present some assessment of the sensitivity of the model results presented (i.e., areal extent of predicted instability) to unknown values introduced to or inferred from the model, like the ice-debris friction coefficient or debris hydraulic conductivity. These could significantly change the results.

Thanks for suggesting the sensitivity assessment it was remiss of us to not include one. We now include a sensitivity test routine in which each of the parameters is perturbed in turn, and we include the description and findings of this in our methods and discussion as follows:

Additional section in Methods:

"In order to assess the robustness of the slope stability model, sensitivity tests were carried out for each study area, in which key variables of the slope stability model (ratio of densities of water to debris; saturated hydraulic conductivity; debris-ice interface friction coefficient; debris thickness and calculated daily melt rate) were perturbed, one at a time, by ± 10 %. The percentage of the study area classified as unstable, as well as percentage change from that study area's areal percentage instability (using the best estimate values given above), was recorded for each perturbation."

Additional section in Results and Discussion:

"Perturbing slope stability model input variables by 10% generally resulted in small changes of up to 1% in areal percentage slope instability, indicating the model is relatively robust. However, adjusting the debris-ice friction coefficient by 10% caused relatively large changes of up to 9%. Increasing melt rate and the density of water to the density of wet debris ratio cause areal percentage slope instability to increase. Increasing hydraulic conductivity, the debris-ice friction coefficient, and debris thickness cause areal percentage slope instability to decrease. It is interesting to note that the upglacier study area is most sensitive to input variable perturbation, presumably because debris thickness and therefore melt rate are greatest in the upglacier study area."

Figure 2's caption and the text on line 109 indicate that there should be a panel (b) for Figure 2, but none appears in the copy of the manuscript I've seen.

Thanks, we now include the photograph for Figure 2b as originally intended.

[Figure]

Finally, we would like to point out three changes that have been made further to those requested in the reviews.

1) While doing the additional sensitivity tests on the slope stability model suggested by reviewer 1, we noticed a coding error causing areal percentage slope tability/instability excluding ponds and ice cliffs to be wrong. We have adjusted the values accordingly in the manuscript and figures. This does not affect the conclusions of the paper, but rather strengthens our argument that relatively large areas of the debris surface are unstable, on the basis that the values that exclude ponds and ice cliffs are now more similar to those that include ponds and ice cliffs.
This led to a change in the text as follows: "Slope stability modelling suggests that, under mid-August ablation conditions, the percentage of the debris-covered area interpreted as potentially unstable for the three study areas of Ngozumpa Glacier is between 13 and 34% including ponds and ice cliffs, and between 12 and 22% 10 and 32% if ponds and ice cliffs are excluded (Fig. 9)."
2) We also noticed that we had used the incorrect colour map in Figure 9d and this has also been corrected in the revised manuscript.
3) The reference to Del Gobbo (2017) was previously missing from the reference list, but has been added now.

---

## Author Comment (AC2) · 7 Sep 2018

**Response to reviewer 2:**

We would like to thanks reviewer 2 for a thorough review of our paper that identified a number of inconsistencies and ambiguities that we are happy to be able to clear up.

Our responses are given in blue below and highlighted in the revised manuscript using track changes.

**Main comments:**

1) Author have analyzed debris thickness frequency which are measured by GPR, theodolite, SfM-MVS and excavation. I can imagine point measurements of debris thickness were carried out by theodolite, SfM-MVS and excavation. But, this paper have no information about the GPR. GPR also measure DT at each point or at some ranges in horizontal ( depend on the debris thickness??)?

The GPR does indeed sample a footprint rather than a single point, that is partially dependent on instrument geometry and the distance to the reflector. We used the GPR in a continuous sense to collect profiles of data, and then picked the reflector surface from this. Datapoints were extracted from this picked surface for subsequent analysis. We now add that:

"Debris thickness data was extracted from the picked ice surface at approximately 0.02 m ground spacing for subsequent data analysis."

2) You have classified stable and unstable debris-covered area at just 27 degree in slope gradient. But, usually, steep slope have low accuracy in satellite- or SfM-DTM. Margin site might have no oversteepened grids if you change the critical value of slope.

The classification was based on the available data for our study site. Certainly a wider sampling of debris thickness could lead to adjustment of our classification of stable and unstable ground, but such data is not available. Our DTM from the Pleaides imagery resolves the steep slopes in our study area adequately, but we agree once could encounter issues if the available DTM is of insufficient quality.

We have now added a sensitivity study to the results of the stability modelling and find that:

"Perturbing slope stability model input variables by 10% generally resulted in small changes of up to 1% in areal percentage slope instability, indicating the model is relatively robust. However, adjusting the debris-ice friction coefficient by 10% caused relatively large changes of up to 9%. Increasing melt rate and the density of water to the density of wet debris ratio cause areal percentage slope instability to increase. Increasing hydraulic conductivity, the debris-ice friction coefficient, and debris thickness cause areal percentage slope instability to decrease. It is interesting to note that the Upglacier study area is most sensitive to input variable perturbation, presumably because debris is thinner and therefore melt rate are greatest in the Upglacier study area."

**Specific comments:**

Please consolidate the location name. Even all expressions are summarized in

Line 199-201, different expressions at each figure are not easy to understand for readers. You have wrote 'Gokyo' and 'Margin' in Fig. 2, 4, 6c, 6d and 9, but expressed by distance from terminus in Fig. 5. Please check other part in the manuscript. 'lower' 'middle' and 'upper' were also found in Line 382.

We have checked the manuscript and now use 'Upglacier', 'Gokyo' and 'Margin' to refer to the study sites shown in Figure 1 consistently throughout.

L120 Please add information of altitudes at each three site.

We added this information to the caption of Figure 2: "(a) Ngozumpa glacier showing the key study areas, ~7, 2 and 1 km from the glacier terminus at elevations of 4870, 4750 and 4740 m a.s.l. respectively"

L163 Author have wrote that 'McCarthy et al (2017) and range from 0.14-0.83 m, generally increasing with debris thickness' How much minimum thickness can be detect by GPR? Author have depicted Fig. 5, percentage frequency histgrams of debris thickness in 0.05 m. Thin debris are important for following analysis.

The range 0.14-0.83 m, refers to the uncertainty associated with derived debris thickness, following the error propagation described in McCarthy and others 2017. Minimum detectable debris thickness depends on the GPR operating frequency and 'transmitter blanking effects'. We now add in the methods section that: "According to McCarthy et al (2017), transmitter blanking is limited to one wavelength below the surface and so minimum detectable debris thickness is roughly equal to the ratio of debris wave speed to radar frequency. In our case this would imply minimum detectable debris thickness of 0.27 m with the 600 MHz antenna and 0.80 m with the 200 MHz antenna."

Percentage frequency in Figure 5 is presented in 0.1 m bins. This has now been corrected in the figure caption. This bin interval was chosen to adequately display the data from all studies.

L175 '4.2 Ablation modelling' » It seems that temperature difference due to elevation difference at each three site have not been considered in this calculation. I think it is not necessary to consider the temperature difference, because the target of this calculation is to indicate the effect of debris thickness variability on ablation. But, you have to write that you have assumed that temperature were same with the Pyramids (?) at all sites.

Correct. As the modelling is idealized in any case we did not account for the elevation between our study sites. We have tried to clarify this in the text and now state: "The model does not account for variability in surface energy receipts due to local topoclimate, or the effects of spatially or temporally variable debris properties other than thickness, and the chosen input properties are only approximate. However, this does not preclude its illustrative use in investigating the influence of variable debris thickness on calculated ablation rate. Ablation modelling was carried out using the same forcing data varying only the local debris thickness information determined at: (i) the Margin study site ~1km from the glacier terminus, (ii) the main Gokyo study site ~2 km from the terminus, both measured by GPR in 2016, and (iii) the Upglacier study site ~7 km from the terminus, measured by theodolite survey in 2001 (Fig. 2)."

L239-240 There are no information of data source of air photographs taken in 1984.

Added reference to Washburn 81989) which provides details of the Swissair image flight.

Washburn, B.: Mapping Mount Everest, Bull. Am. Acad. Arts Sci., 42(7), 29–44, 1989.

L299-301 'the debris was generally too thick. This means there is the possibility of a slight thin bias in the data. However, penetration depth was often greater than 7 m, which is likely near the maximum debris thickness.' » Those sentences have been written subjectively without support information. Authors cannot declare without some references.

Thank you, we agree this type of claim should be avoided. We now rephrased this as: "For those radargrams in which the ice surface was not easily identifiable, the debris appeared to be too thick to detect. While this means there is the possibility of a slight thin bias in the data, it is reasonable to assume the impact is minimal because penetration depths exceed the thickness of any supraglacial debris exposures observed in the field (Nicholson and Benn, 2012; Nicholson and Mertes, 2017)."

L307 'see Section 5.3 and Fig. 6' » 'see Section 5.3 and Fig. 7'?

Thank you. This has been corrected to Fig 7.

L343 I cannot find Fig. 2b.

Thanks, we now include the photograph for Figure 2b as originally intended.

L354 If you have assumed that temperature difference depending on the altitude were not considered for the estimate of ablation, you have to add the information here. If you take into account the temperature difference, you have to add altitude at each three sites and temperature lapse rate.

We used the same meteorological forcing, as described in the methods section. To reiterate this point we now write: "The ablation calculated for typical August conditions at the pyramid weather station using the mean debris thickness at the Margin, Gokyo and Upglacier sites …"

L355 and 362 The unit of vertical axis in Fig. 6c were mm day-1. but, 'm' in the manuscript. If both values indicate same things (I believe this), please consolidate. This figure and calculated values are very important in this paper.

Our original intent was that the monthly total lowering is easier to vizualise than a lowering in mm/day. In order to maintain consistence we now provide both metrics in the text as follows: "The ablation calculated for typical August conditions at the pyramid weather station using the mean debris thickness at the Margin, Gokyo and Upglacier sites was 2.2, 3.6 and 10.5 mm day-1 (Fig. 6c), totalling  0.07, 0.11 and 0.33 m of ice surface lowering over the month respectively. This agrees with the general expected patterns of ablation gradient reversal towards the terminus of a debris-covered glacier (e.g. Benn and Lehmkuhl, 2000; Bolch et al., 2008; Benn et al., 2017). Accounting for the percentage frequency distribution of debris thickness at the Margin, Gokyo and Upglacier sites increased the surface lowering rate to 2.5, 5.2 and 15.0 mm day-1, giving monthly total surface lowering of 0.08, 0.16 and 0.46 m respectively."

L360 ' 1, 3 and 7 km respectively.' »' 1, 2 and 7 km respectively.' ???

This is now rephrased to keep with the site naming convention: "The ablation calculated for typical August conditions at the pyramid weather station using the mean debris thickness for at the Margin, Gokyo and Upglacier sites totalled 0.07, 0.11 and 0.32 m of ice surface lowering over the month respectively. This agrees with the general expected patterns of ablation gradient reversal towards the terminus of a debris-covered glacier (e.g. Benn and Lehmkuhl, 2000; Bolch et al., 2008; Benn et al., 2017). Accounting for the percentage frequency distribution of debris thickness at the Margin, Gokyo and Upglacier sites increased the monthly total surface lowering due to ablation to 0.08, 0.16 and 0.46 mrespectively."

L370-373 These sentence indicates very significant things to estimate ablation under debris-layer based on mean debris thickness. I recommend that each percent of debris thickness frequency between 0-0.5 m in debris thickness and each calculated ablation ratio between 0-0.5 m in debris thickness at each three site should be shown in the text.

We initially avoided highlighting the exact values resulting from our model as we intend it to be illustrative. The exact values will depend on the local Østrem curve and local debris thickness distribution.

For the mid-bin values of the distributions shown in Figure 5 the ration of debris covered to clean ice ablation are as follows

| $h_d$ bin (m) | 0.0 - 0.1 | 0.1 - 0.2 | 0.2 - 0.3 | 0.3 - 0.4 | 0.4 - 0.5 |
|---|---|---|---|---|---|
| Margin (GPR) | 0 | 0 | 0 | 0 | 0 |
| Gokyo (GPR) | 0 | 0 | 0 | 1 | 2 |
| Upglacier (survey) | 1 | 16 | 22 | 11 | 11 |
| $h_d$ bin midpoint (m) | 0.05 | 0.15 | 0.25 | 0.35 | 0.45 |
| Ablation ratio | 0.92 | 0.59 | 0.44 | 0.34 | 0.29 |

We can add this information into the manuscript if deemed essential but prefer not to as its specific to the illustrative cases given in the paper.

L389 'Visual inspection of the radargrams indicates that the thinnest debris cover occurs on steep slopes (Fig. 7a and b).' » I cannot agree with this sentence. For me, by visual inspection, it seems that depression of ice surface are filled with debris, as a result, debris surface have flatter features than that of ice surface under the debris. All example show such tendency in Fig. 7.

Yes, this is a good way of expressing the general pattern and we now write: 2 Visual inspection of the radargrams indicates that the thickest debris is found filling depressions in the underlying ice surface, and thinner debris is more commonly seen overlying steeper ice surfaces (Fig. 7a and b)."

Line394 'the debris surface is approximately parallel to the ice surface,' » This sentence does not conflict with previous sentence 'Visual inspection of the radargrams indicates that the thinnest debris cover occurs on steep slopes' at Line 389 ??

We now write: "On steeper slopes where the debris surface is approximately parallel to the ice surface, this appears to be a characteristic of debris covers at or near the limits of gravitational instability."

L390 '(Fig. 7a and b).'» '(Fig. 7a and c).'???

Changed to Fig 7a.

L397 'Modelled surface flowpaths (Fig. 7b) cross-cut the GPR transects where these depressions are located, indicating that they were likely incised by meltwater.' » I'm confused when I checked the cross section of Fig. 7c and f. You have calculated the flowpath based on DEM from Pleiades. But, the surface features indicated by GPR cross section do not represent cross section of flowpath (no depression). The surface elevation of debris is not true?

The surface elevation of the profiles shown is topographically corrected. However, as the GPR profiles only show the surface in two dimensions, whereas upstream contributing area, which was used to generate the flow paths, was calculated using the 3D DTM, there is not necessarily a clear depression across the profile associated with each flowpath.

L417 and Fig. 7h » How did you detect the location of former pond? 'Former pond' in Fig. 7h means pond in 1984?

The former pond locations are indicated from the air photograph and satellite image mapping, but also based on field observations that the radar line location here crossed well sorted, stratified fines, indicating lacustrine deposition.

We now say: "Since 1984, the existence of supraglacial ponds within the Gokyo study area is likely to have affected two areas of radar transects: Several transects towards the north of the Gokyo study area, which may have been partially affected by lakes in 2012 and 2014, and a single transect towards the east of the Gokyo study area, which crossed clearly lacustrine surface deposits was partially affected by lakes in all the sampled years except 2014 and 2016 (Fig. 4)."

L434-435 'Binning the thickness data with respect to slope indicates a step decrease in debris thickness above surface slope angles of around 20-23 Í ˛e (Fig. 8a).' » I recommend that difference between debris thickness at steep slope and those at gentle slope were significant or not statistically.

We now added: "...slope indicates a non-statistically significant step decrease ..."

L434- As I wrote at Line 394, it seems that depression of ice surface are filled with debris, as a result, debris surface have flatter features than that of ice surface under the debris from Fig. 7 by visual inspection. Then, ice surface (not debris surface) curvature might have relation with debris-thickness. Probably I think some relation can be found between ice surface curvature and debris-thickness by optimizing the scale of curvature. But, it's not your purpose, and the analysis is limited at the GPR survey lines.

We agree, but we can't calculate ice surface curvature from two-dimensional GPR profiles, and the aim of this part of the analysis was to see if a viable relationship existed between debris thickness and (relatively) readily determined surface properties. We modified this sentence slightly to clarify this: "The debris thickness sampled with GPR in this study does not show distinct relations with surface slope, aspect or curvature, that could be readily extracted from glacier surface terrain models (Fig. 8a, b, c)."

L448 'This effect is observable in global radiation data (Fig. 8d).'» If you add calculated solar radiation at the top of the atmosphere in Fig. 8d, it is easy to understand.

We have added this into the figure.

Fig. 4 Shore lines of ponds are unclear for me because they are overlapping. I recommend that only shore lines are depicted in this figure. And add the information of the background images.

We prefer to show the GPR lines in the context of the ponds. The background images is given in the caption as the hillshade from the Pleiades DTM, and the images from which the lakes are mapped are described in the text: "The recent history of ponded water on the parts of the glacier surface sampled by the radar transects was mapped using air photographs from 1984 (see Washburn, 1989 for details), and seven cloud-free optical satellite images spanning 2008-2016. The satellite images consisted of six Digital Globe images, and one CNES/Astrium image, all obtained via Google Earth, and the optical image from the 2016 Pleiades acquisition used to generate the DTM." We feel this adequately described the figure source information and would prefer to keep it as it is.

Fig. 5 There is no information about the intervals of debris thickness in d-f. And please add the information that 'Dashed vertical lines indicate mean debris thickness' or something.

Caption now reads: "Figure 5: Percentage frequency histograms of debris thickness (hd) in 0.1 m intervals, and mean debris thickness as vertical dashed lines for (a) the Ngozumpa Margin study site; (b) the Ngozumpa Gokyo study site;(c) the Ngozumpa Upglacier study site; (d) over the lower tongue of Lirung glacier in central Nepal; (e) across the debris covered ablation area of Suldenferner/Ghiacciaio de Solda in the Italian Alps and (f) the medial moraine of Haut Glacier d'Arolla in the Swiss Alps. Measurement methods are GPR (black); theodolite surveys (blue); Structure from Motion (SfM-MVS) photographic terrain model (green) and excavation of pits (red). Note that axes vary between sites, and summary statistics of these distributions are in Table 2."

Fig. 7 Locations of Fig. 7b, f, h are required.

As the figure is already quite busy, and the specific locations of each radar transect shown are not as relevant as their relation to the other features we prefer to leave this figure as it is.

Table 2 I recommend that percentage of thin debris thickness frequency are necessary in Table 2. For example, the range between 0-0.25 m and 0.25-0.5 m in debris thickness. Because, the thin debris layer effect the bias of ablation and surface temperature

For the mid-bin values of the distributions shown in Figure 5 the ration of debris covered to clean ice ablation are as follows

| $h_d$ bin (m) | 0.0 - 0.1 | 0.1 - 0.2 | 0.2 - 0.3 | 0.3 - 0.4 | 0.4 - 0.5 |
|---|---|---|---|---|---|
| Margin (GPR) | 0 | 0 | 0 | 0 | 0 |
| Gokyo (GPR) | 0 | 0 | 0 | 1 | 2 |
| Upglacier (survey) | 1 | 16 | 22 | 11 | 11 |
| $h_d$ bin midpoint (m) | 0.05 | 0.15 | 0.25 | 0.35 | 0.45 |
| Ablation ratio | 0.92 | 0.59 | 0.44 | 0.34 | 0.29 |

We can add this information into the manuscript if deemed essential but prefer not to as its specific to the illustrative cases given in the paper.

Finally, we would like to point out three changes that have been made further to those requested in the reviews.

1) While doing the additional sensitivity tests on the slope stability model suggested by reviewer 1, we noticed a coding error causing areal percentage slope tability/instability excluding ponds and ice cliffs to be wrong. We have adjusted the values accordingly in the manuscript and figures. This does not affect the conclusions of the paper, but rather strengthens our argument that relatively large areas of the debris surface are unstable, on the basis that the values that exclude ponds and ice cliffs are now more similar to those that include ponds and ice cliffs.
   This led to a change in the text as follows: "Slope stability modelling suggests that, under mid-August ablation conditions, the percentage of the debris-covered area interpreted as potentially unstable for the three study areas of Ngozumpa Glacier is between 13 and 34% including ponds and ice cliffs, and between 12 and 22% 10 and 32% if ponds and ice cliffs are excluded (Fig. 9)."
2) We also noticed that we had used the incorrect colour map in Figure 9d and this has also been corrected in the revised manuscript.
3) The reference to Del Gobbo (2017) was previously missing from the reference list, but has been added now.

---

## Author Response (AR1)

Dear Valentina and reviewers,

We would like to thank the reviewers for their valuable input to improving our manuscript.

The point by point responses to the reviewer comments were made in the individual replies we posted as supplements in the discussion (https://www.the-cryosphere-discuss.net/tc-2018-83/). Accordingly, rather than repeating those here we simply attach the revised manuscript including all changes highlighted.

We would like to additionally point out three changes that have been made further to those requested in the reviews.

1) While doing the additional sensitivity tests on the slope stability model suggested by reviewer 1, we noticed a coding error causing areal percentage slope stability/instability excluding ponds and ice cliffs to be wrong. We have adjusted the values accordingly in the manuscript and figures. This does not affect the conclusions of the paper, but rather strengthens our argument that relatively large areas of the debris surface are unstable, on the basis that the values that exclude ponds and ice cliffs are now more similar to those that include ponds and ice cliffs.

This led to a change in the text as follows: "Slope stability modelling suggests that, under mid-August ablation conditions, the percentage of the debris-covered area interpreted as potentially unstable for the three study areas of Ngozumpa Glacier is between 13 and 34% including ponds and ice cliffs, and between 12 and 22% 10 and 32% if ponds and ice cliffs are excluded (Fig. 9)."

2) We also noticed that we had used the incorrect colour map in Figure 9d and this has also been corrected in the revised manuscript.

3) The reference to Del Gobbo (2017) was previously missing from the reference list, but has been added now.

[revised manuscript text omitted]